# Dental Age Estimation Standards for Hispanic Children and Adolescents in California

**Adriana Ustarez [1], Daniela Rodrigues Silva [1], Graham Roberts [2] and Jayakumar Jayaraman [3],***

[1] Section of Pediatric Dentistry, University of California at Los Angeles, Los Angeles, CA 90024, USA
[2] Department of Orthodontics, King's College London, Guy's Hospital, London SE1 9RT, UK
[3] Department of Pediatric Dentistry, Virginia Commonwealth University School of Dentistry, Richmond, VA 23298, USA
* Correspondence: jayakumar83@hotmail.com; Tel.: +1-804-828-2362

**Abstract:** *Background:* In recent years, cross-border migrations have resulted in an increase in the number of unaccompanied children apprehended at the United States border, particularly in the state of California. The assessment of the chronological age of a child, in many instances, determines the type of services rendered within the medico-legal system. Age can be determined by using population-specific reference standards, preferably within a geographical area. However, such standards are not available for Hispanic children living in California. Aim: To present new standards by developing and validating a reference data set for dental age estimation in Hispanic children in California. *Methods:* For the reference dataset, a total of 705 dental panoramic radiographs of healthy children aged 7.00 to 13.99 years belonging to Hispanic ethnicity in California were obtained from the archives of a teaching hospital. All permanent teeth on the left side were scored in automated software, and the average at assessment was calculated for each stage of dental development. For the validation dataset, 133 radiographs that were not part of the reference dataset were obtained based on the above criteria. The difference between the chronological age (CA) and dental age (DA) estimated using the California Hispanic reference dataset was assessed using a paired *t*-test with a statistical significance of $p < 0.05$. *Results:* The overall difference between the chronological age and dental age (CA-DA) was 0.03 years (1.56 weeks) for females and $-0.10$ years ($-5.26$ weeks) for males, and the difference was not statistically significant for children aged 8.00 to 12.99 years ($p > 0.05$). *Conclusions:* The newly constructed dental reference data can be recommended for age estimation in children belonging to Hispanic ethnicity in California.

**Keywords:** forensic science; legal medicine; immigration; panoramic radiographs; United States; reference data set

## 1. Introduction

Various methods exist for the determination of age, including sexual maturation, psychological development, and fusion of the epiphyseal cartilages in the hand, wrist, and the sternoclavicular joint [1]. Although dental development has been used for many centuries for the estimation of chronological age, it has been established that the estimation of chronological age is reliable compared to other methods. Several methods have been developed to analyze and interpret dental development and subsequently assist in dental age estimation. One of the earlier methods, although not indicated for age estimation, was developed by Demirjian and co-workers [2], but it has been shown to consistently overestimate age in global population groups and appears to be ethnic-group specific [3,4].

The Demirjian method of staging permanent tooth development is known in the scientific world as a simple and reproducible system with good intra-examiner reliability for the estimation of chronological age [5]. This method uses an eight-stage system whereby each tooth in the mandibular and maxillary left side is evaluated and assigned a stage, for example, Stage A at the first sign of calcification through H at the closure of the

apex [2]. The Dental Age Research London Information Group (DARLInG, London, United Kingdom) has used this system to develop reference data sets of tooth development in many ethnic groups [5]. This method has been well reported and has been shown to accurately estimate the age of subjects using ethnic-specific population groups in Afro-Caribbean [6], Caucasian [7–9], Chinese [10], Kuwaiti [11], and Maltese populations [12].

In recent years, cross-border migrations have resulted in an increase in the number of unaccompanied children apprehended at the United States border, particularly in the state of California. In San Diego alone, a 25.7% increase in the number of apprehended unaccompanied children and single minors was reported between 2021 and 2022. This amounted to 98,654 children in the 2022 financial year to date (FYTD) compared to 78,459 children in 2021 [13]. Based on the US Census Bureau, Hispanics are the largest minority group in the state of California and is expected to grow at a rate of over 93% from 2016 to 2060 [14]. Not only does the Los Angeles Metropolitan area have the largest concentration of Hispanics, 73.7% of the children enrolled in the Los Angeles Unified School District are of Hispanic origin [15]. Most of the children seeking asylum do not possess authentic birth documentations, and falsification of age has been reported considering the legal entitlements provided to a child, i.e., below 18 years of age [16]. The assessment of the chronological age of a child, in many instances, determines the type of services rendered within the medico-legal system. This includes counseling, foster care, education, shelter, provision of medical and dental care, etc. In a similar matter, dental age is also regarded as an important factor in the field of orthodontics and pediatric dentistry since treatment methods and planning are often specifically based on the child's dental age. It has been well documented that chronological age can be determined by using population-specific reference standards developed from identifiable human groups, preferably within a geographical area [17]. Although different ethnicities have been studied, to date, only one study has used the Demirjian's staging criteria to assess the stage of development of third molars in the Hispanic population [18]. A recent study reported dental reference standards based on two staging systems in Hispanic children in Texas [19]. However, no such standards are available for Hispanic children living in California. Hence, the aim of this study was to develop and validate a reference data set for dental age estimation of Hispanic children in California.

## 2. Materials and Methods

### 2.1. Sample Population

Electronic charts within the Sections of Pediatric Dentistry and Orthodontics at the University of California Los Angeles (UCLA) School of Dentistry, California, United States were the source of dental panoramic radiographs used in this study. The radiographs used in both the construction and validation phases of this study were previously taken for diagnostic purposes and were re-used for this study. The inclusion criteria were healthy children of Hispanic ethnicity without anomalies that might affect dental development. The exclusion criteria were panoramic radiographs of poor diagnostic quality that prevent scoring the tooth developmental stages. The ethnicity determination was made from the patient's parent report on their registration form. In addition, it has been recorded that both parents were residents of California, and the child was born in the same state. For the reference data set, samples were obtained from patients attending either of these two department clinics from 1995 to 2017. The subjects were categorized based on age and gender. For the reference data set, a total of 705 records were assessed, and approximately 50 panoramic radiographs per age and per gender were studied. If there were multiple panoramic radiographs for the same child, only one was included in the study. Ethical approval for this study was obtained from the Institutional Review Board at the University of California Los Angeles. (IRB # 17-001225). In total, the reference data set comprised a total of 705 subjects—355 males and 350 females whose ages ranged from 7.00 to 13.99 years. The mean age of the females and males in the reference dataset was 10.51 and 10.50 years, respectively. The validation set comprised 66 females and 67 males in the same age range as the reference data set. Table 1 shows the sample size distribution based on gender and age.

**Table 1.** Distribution of children by age and sex utilized in the construction of the reference data set.

| Age (Years) | Males | | | Females | | |
|---|---|---|---|---|---|---|
| | *n* | Mean Age | SD | *n* | Mean Age | SD |
| 7.00–7.99 | 51 | 7.52 | 0.29 | 50 | 7.55 | 0.30 |
| 8.00–8.99 | 52 | 8.54 | 0.28 | 50 | 8.52 | 0.29 |
| 9.00–9.99 | 51 | 9.51 | 0.29 | 51 | 9.51 | 0.28 |
| 10.00–10.99 | 52 | 10.45 | 0.28 | 50 | 10.49 | 0.29 |
| 11.00–11.99 | 50 | 11.47 | 0.26 | 50 | 11.56 | 0.24 |
| 12.00–12.99 | 48 | 12.53 | 0.29 | 49 | 12.48 | 0.31 |
| 13.00–13.99 | 51 | 13.52 | 0.26 | 50 | 13.51 | 0.32 |
| **Total** | | | | | | |
| 7.00–13.99 | 355 | 10.50 | 0.28 | 350 | 10.51 | 0.29 |

### 2.2. Data Processing

All patient's identifying details were eliminated by using a unique designated identification number. All data were stored in an independent digital memory device. A calibrated examiner (AU) assessed the panoramic radiographs. The examiner is a pediatric dentistry resident at the UCLA School of Dentistry. To assess the inter examiner reliabilities, 20 unidentified panoramic radiographs from a different institution were assessed by two examiners (AU, DRS) independently, and the examiners were calibrated by a third examiner (JJ). The images for this study were accessed from the patient's electronic record and viewed via the XDR software on a widescreen monitor at standard magnification without the use of any editing software. Each panoramic radiograph was scored. The scoring was completed by assigning a stage from A to H (Figure 1) for each individual tooth on the left side (upper and lower jaws) and the third molars on the right side, making a total of 18 teeth. Stage A represents initial calcification and Stage H indicates completion of root development [2].

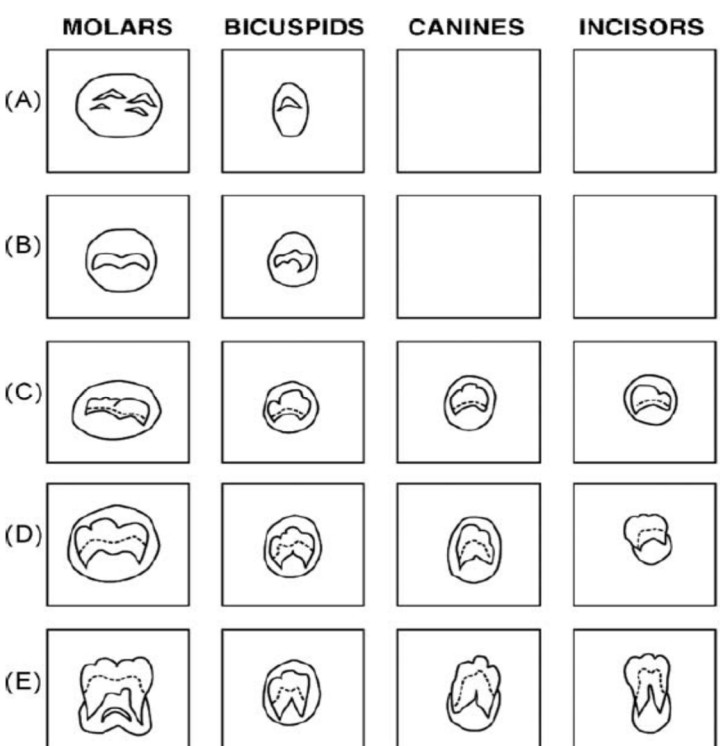

**Figure 1.** *Cont.*

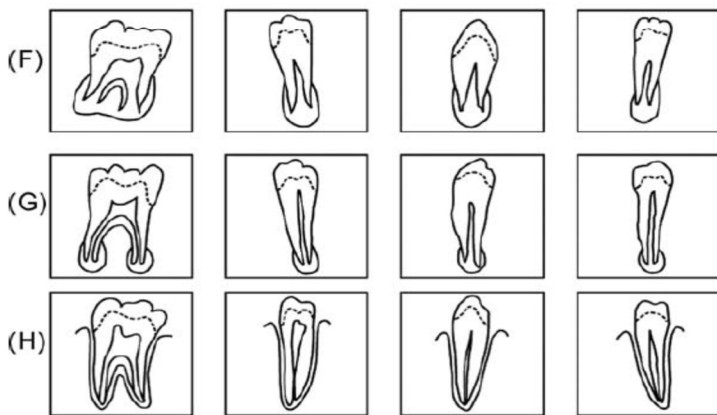

**Figure 1.** Classification of tooth developmental stages for single and multi-rooted teeth (Demirjian et al., 1973) [2].

### 2.3. Construction of Hispanic Reference Data Set (RDS)

Since all the scoring was completed in paper form, data were then transferred into a Microsoft Access database (Microsoft Corp, Redmond, WA, USA). Data, including date of birth, date of radiograph exposure, gender, and ethnicity, along with tooth development stage scoring, were recorded in the Microsoft Access database designed by the Dental Age London Information Group (DARLInG), London, United Kingdom [5]. This software calculates the chronological age in decimal years of the age at assessment (AaA) for each of the tooth development stages (TDS). The final output in the form of average age was obtained for each TDS for males and females separately. From these data, the reference data set (RDS) was developed.

### 2.4. Sample Population for Validation Set (VS)

The validation set (VS) comprised 133 panoramic radiographs of healthy, non-syndromic Hispanic subjects who attended UCLA Departments of Pediatrics and Orthodontics between 1995 and 2017 and were not a part of the reference dataset. Approximately ten subjects per age per gender were chosen randomly based on a computer-generated number. If there were multiple panoramic radiographs for the same child, only one was included in the validation set. Each panoramic radiograph was scored by two trained and calibrated examiners simultaneously (AA,DS). If there was disagreement on the assigned score for a tooth, then a consensus was reached prior to determining the final score for that tooth with the help of the third examiner (JJ).

### 2.5. Calculation of Chronological Age and Dental Age

Chronological age was calculated from the difference between the date of birth and the date when the radiograph was taken. The details of each subject were entered into Microsoft Excel (Microsoft Corp, Redmond, WA, USA). Dental age was calculated from the number (n), mean (x), and standard deviation (sd) corresponding to each tooth based on the RDS.

### 2.6. Comparison of Chronological Age and Dental Age

Inter and intra examiner repeatability for assessing the tooth development stage for each tooth in the validation set was tested with a weighted Kappa analysis [20]. SPSS software (Version 21, IBM Inc, Armonk, NY, USA) was used for all statistical computations. Statistical significance was set at $p < 0.05$, and a paired $t$-test was used to compare the difference between the chronological age (CA) and the dental age (DA) for males and females separately for each age range from 7.00 to 13.99 years.

## 3. Results

### 3.1. Examiner Reliability

The Kappa values for the intra-agreement examiner score for the first examiner (AU) was 0.94 at the start of RDS development and 0.87 halfway into the RDS development process ($p < 0.001$). Both scores indicated "almost perfect" agreement. The inter-agreement examiner score between the first and second examiner (AU,DS) was 0.71, indicating that the observed scores were "good" ($p < 0.001$) [20].

### 3.2. Chronological Age vs. Dental Age

The validated Hispanic RDS based on dental maturation accurately estimated the chronological age for males between ages 8.00 to 12.99 years, and no difference was observed between the chronological and dental age ($p > 0.05$). The overall difference between the chronological and dental age (CA-DA) for all females was −0.10 years (−5.21 weeks) and for males, it was 0.03 years (1.56 weeks). The validated Hispanic RDS on dental maturation accurately estimated the chronological age for females between ages 9.00 and 12.99 years. The most accurate estimates were observed from 12.00 to 12.99 years in the males and females at −0.06 years and 0.13 years, respectively ($p > 0.05$). The exact differences in each age group and gender are summarized in Table 2.

**Table 2.** Difference between the chronological age (CA) and dental age (DA) in Hispanic males and females estimated from the Hispanic reference dataset.

| Age (Years) | Males | | | | Females | | | |
|---|---|---|---|---|---|---|---|---|
| | CA | DA | CA-DA | *p*-Value | CA | DA | CA-DA | *p*-Value ^ |
| 7.00–7.99 | 7.49 | 7.76 | −0.27 | 0.018 * | 7.57 | 8.78 | −1.22 | 0.001 * |
| 8.00–8.99 | 8.50 | 8.37 | 0.13 | 0.459 | 8.31 | 9.09 | −0.78 | 0.001 * |
| 9.00–9.99 | 9.57 | 9.86 | 0.13 | 0.201 | 9.41 | 9.67 | −0.26 | 0.074 |
| 10.00–10.99 | 10.44 | 10.64 | −0.29 | 0.399 | 10.61 | 10.82 | −0.22 | 0.372 |
| 11.00–11.99 | 11.47 | 11.52 | −0.20 | 0.850 | 11.48 | 11.1 | 0.39 | 0.083 |
| 12.00–12.99 | 12.34 | 12.07 | −0.06 | 0.119 | 12.45 | 12.31 | 0.13 | 0.162 |
| 13.00–13.99 | 13.46 | 12.88 | 0.26 | 0.003 * | 13.51 | 12.58 | 0.93 | 0.001 * |
| **Total** | 10.47 | 10.45 | 0.03 | 0.714 | 10.63 | 10.74 | −0.10 | 0.284 |

CA—chronological age, DA—dental age, ˆ Paired *t*-test, * statistically significant $p < 0.05$.

### 3.3. Working Example

To demonstrate the dental age estimation calculation, we have presented a working example of a male child of Hispanic ethnicity. The date of birth (DOB) of the child was 29 July 2007, and the date of exposure of the radiograph (DOR) was 6 June 2017. The chronological age was determined as 9.92 years based on the simple formula in Microsoft Excel ((DOR-DOB)/365.25). Figure 2 shows a dental panoramic radiograph of this child with multiple permanent teeth developing. The stage of development for each tooth and the corresponding scores obtained from the Hispanic dental reference dataset are shown in Table 3. Based on mean age at assessment (AaA) of fourteen developing teeth, the dental age (DA) was calculated as 9.80 years. The difference between the chronological and dental age (CA-DA) was 0.12 years.

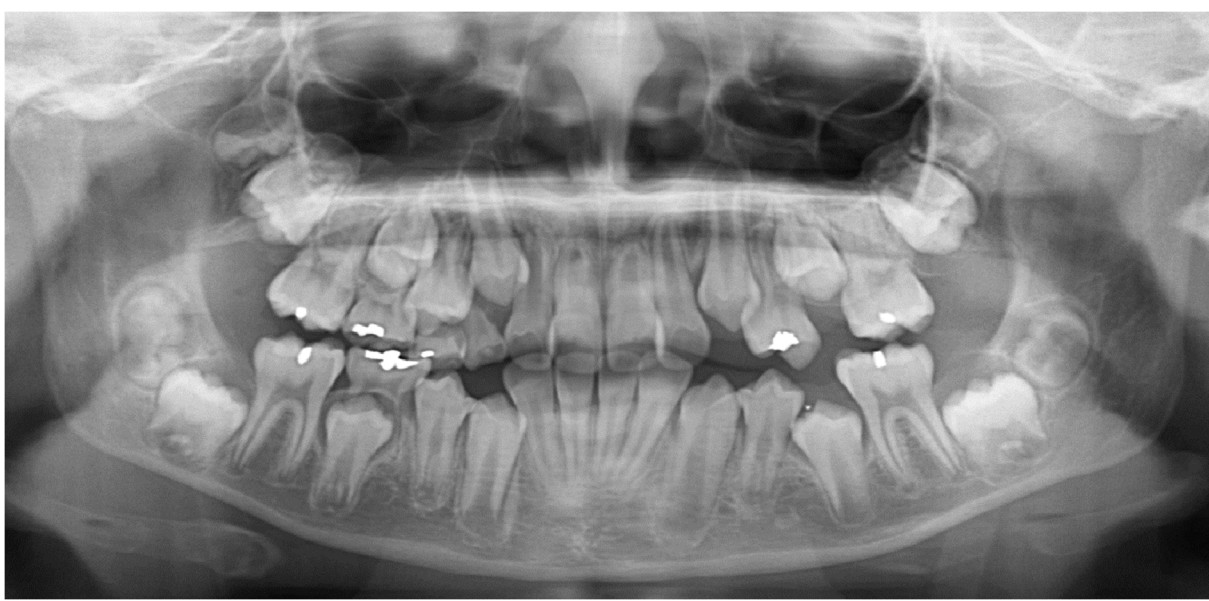

**Figure 2.** Dental panoramic radiograph of a male subject of Hispanic ethnicity aged 10.20 years.

**Table 3.** Stage of dental development for the subject presented in Figure 2. The scores corresponding to each tooth were obtained from the Hispanic reference data set for males.

| Tooth | Stage | n-Tds | x-Tds | sd-Tds |
|---|---|---|---|---|
| *Maxillary* | | | | |
| **UL1** | H | — | — | — |
| **UL2** | G | 94 | 9.81 | 1.00 |
| **UL3** | F | 188 | 9.70 | 1.34 |
| **UL4** | F | 99 | 10.06 | 1.00 |
| **UL5** | E | 131 | 9.00 | 1.03 |
| **UL6** | H | — | — | — |
| **UL7** | D | 167 | 8.33 | 0.96 |
| **UL8** | B | 74 | 10.59 | 1.28 |
| **UR8** | B | 80 | 10.72 | 1.36 |
| *Mandibular* | | | | |
| **LL1** | H | — | — | — |
| **LL2** | H | — | — | — |
| **LL3** | F | 180 | 9.65 | 1.31 |
| **LL4** | F | 99 | 10.06 | 0.98 |
| **LL5** | F | 99 | 10.47 | 1.05 |
| **LL6** | G | 132 | 8.09 | 0.99 |
| **LL7** | E | 96 | 9.49 | 1.03 |
| **LL8** | B | 62 | 10.67 | 1.30 |
| **LR8** | B | 64 | 10.62 | 1.18 |
| **Dental Age** | — | — | 9.80 | — |

## 4. Discussion

To our understanding, this is the first ever dental reference dataset for Hispanic children based in California. Although no difference was observed between chronological and dental age in most of the age ranges, the dental age of Hispanic males aged 7.00 to 7.99 years and 13.00 to 13.99 years as well as for females aged between the ages of 7.00 to 8.99 and 13.00 to 13.99, respectively, was significantly different to the chronological age. This might be due to a smaller number of representative samples in the tooth development stages in the Hispanic reference dataset corresponding to the outer age ranges, particularly from 6.00 to 6.99 years and 14.00 to 14.99 years. Based on these findings, dental age appeared to be significantly more advanced in the youngest male and female groups. These results could be attributed to multiple factors. The first and perhaps most influential being

that the reference data set did not include the assessment of younger tooth stages. Assessing permanent tooth development for even younger Hispanic children within the reference data set would have allowed for a more comprehensive representation of the variation in dental development. For example, the reference dataset developed by Jayaraman and co-workers studied ages 2 to 24 years in Southern Chinese children [21]. Due to a lack of panoramic radiographs for younger ages, we only studied children with mixed dentition. The University of California Los Angeles is considered as a "safetynet" hospital, and very often, families of low SES revert to safety net healthcare institutions as these are the only sites that accept publicly funded dental benefits. It has been reported that SES can affect dental age estimation [22,23]. One study found that obese Hispanic children demonstrate an advanced dental age maturation of up to 11.7 months [24]. Similar to our study, this study also assessed CA and DA using the Demirjian staging system in the mixed dentition. Although our study did not account for BMI, the likelihood of having obese Hispanic children in this study is likely since one in every six children in the state of California is considered obese [25].

Several studies have reported significant differences in CA and DA estimations in children older than thirteen years of age [26–28]. A recent study that tested the applicability of the London Atlas on Hispanic population found an overestimation of age in 3% of the samples analyzed [29]. The significant difference in CA and DA in both genders in the 13.00 to 13.99 age cohort can be explained because, by this age, there is less variation in dental development. The RDS values of importance for this cohort are essentially limited to the development of the third molars. As it is well known, third molars often are the most variable dentition, and thus, this can limit the accuracy of the chronological age of an individual based on dental age. As in the case of the youngest age groups, it can be argued that assessing older age groups of Hispanic children, teens, and young adults, mainly those of ages fourteen through twenty-four, would allow us to more accurately determine chronological age based on dental age. The overall difference between the chronological and dental age (CA-DA) in the Hispanic children aged 7 to 14 years in the current study was −0.10 years for females (less than one week) and 0.03 years (less than one week) for males. This is similar to the result of a recent study on Hispanic children in Texas, where a difference of 0.07 years for females and 0.03 years for males was reported on children aged 6 to 17 years [19]. This outcome could also be compared with other studies that utilized the same Dental Age Research London Information Group (DARLInG) methodology for the construction and validation of the reference dataset. For example, in the Kuwaiti population, it was 0.14 years for females and −0.33 years for males [30], 0.03 years for females and 0.05 years for males in southern Chinese [10], and 0.03 years for females and 0.05 years for males Afro-Trinidadians [6]. It is to be noted that the ages of subjects included in the above studies were between 2 and 25 years and may not allow direct comparison based on the age of children included in the current study.

It is to be noted that study did not just validate an ethnic-specific reference data set to determine CA based on DA, but the secondary strength is based on the way the RDS was constructed. The limits of agreement for the RDS were set at a maximum of two standard deviations, meaning any outlier value obtained during that stage of the study was not used. By setting up the RDS in this manner, we have ensured a true representation of the population under study. As with any study, however, there were limitations in the current study. Although the "Hispanic" origin was self-reported, the nationality of each subject included in this study was not accounted for. California has a diverse population of Hispanics, most are immigrants from Mexico, but there is also a prominent number of Hondurian, El Salvadorian, Guatemalan, and South Americans. Thus, although we report on a Hispanic RDS, it is important to note that Central American children may exhibit some differences in comparison to South American children. Furthermore, we included panoramic radiographs that had minor dental anomalies, such as canine transpositions and mesiodens. However, a recent study found no significant difference in the dental age of children with and without supernumerary teeth, and hence, it could be argued that the subjects with mesiodens may not

have influenced the outcomes of this study [30]. The findings of this study indicate reliable age estimates for younger children of Hispanic origin in California, and it is recommended to expand this study to include children over 14 years of age.

A limitation of this study is the age of children and adolescents included, which is 7.00 to 13.99 years. This limits the application of age estimation to this specific age range. Due to time constraints and lack of additional resources within the department, we could not include additional age ranges in the study. Future studies will include adolescents and young adults over 14 years, which would make this dataset completely similar to other published studies [5,6,11,19].

## 5. Conclusions

The newly constructed dental reference data set was able to accurately estimate the age of Hispanic children in California, particularly those in the range of 8 to 12 years. The overall difference between the chronological and dental age estimated from the Hispanic reference dataset was found to be minimal within a range of a few weeks. Hence, this reference dataset can be recommended for dental age estimation in children of Hispanic origin in California.

**Author Contributions:** Conceptualization, J.J. and A.U.; methodology, A.U. and D.R.S.; software, J.J., A.U. and D.R.S.; validation, J.J. and G.R.; formal analysis, A.U. and J.J.; investigation, A.U. and J.J.; resources, G.R.; data curation, A.U. and D.R.S.; writing—original draft preparation, A.U. and J.J.; writing—review and editing, J.J. and G.R.; visualization, A.U.; supervision, D.R.S. and J.J.; project administration, D.R.S.; All authors have read and agreed to the published version of the manuscript.

**Funding:** This research received no external funding.

**Institutional Review Board Statement:** This study was conducted in accordance with the Declaration of Helsinki, and ethical approval for this study was obtained from the Institutional Review Board at the University of California Los Angeles (IRB # 17-001225).

**Informed Consent Statement:** Patient consent was waived since all the personal identifiers were masked.

**Data Availability Statement:** Full reference dataset can be obtained by contacting the corresponding author or from the official website of Dental Age London Information Group (DARLInG), London, United Kingdom: www.dentalage.co.uk (accessed on 24 July 2022).

**Conflicts of Interest:** The authors declare no conflict of interest.

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
