# Peer review of "Dental Age Estimation Standards for Hispanic Children and Adolescents in California"

_forensicsci, doi:10.3390/forensicsci2030041_

Round 1

Reviewer 1 Report

Thanks for the manuscript. This is well planned and well written.

1. Justification to include only up to 14 yrs is required. As 16 yrs is considered as a critical age in forensics and legally, this need to be justified. 

2. Inclusion and exclusion criteria for selection of DPT need to be added

3.Comparrison of the findings with other ethnic groups and populations section need to be expanded as this will increase the validity of the study.

Author Response

Reviewer 1

We thank the reviewer for the comment. Please see our response below:

  1. Justification to include only up to 14 yrs is required. As 16 yrs is considered as a critical age in forensics and legally, this need to be justified. 

This is a valid point. This research originated from a student project and due to limited time and lack of resources, we could not include children over 14 years. We have already indicated this as a limitation. We will present the data of 15 to 24 year old to cover adolescents and young adults in a separate paper. The following information has been presented in the “Discussion” section:

“A limitation of this study is the age of children and adolescents included which is 7.00 to 13.99 years. This limits the application of age estimation within to this specific age range. Due to time constraint and lack of additional resources within the department, we could not include additional age ranges in the study. Future studies will include adolescents and young adults over 14 years which would make this dataset complete similar to other published studies [5,6,11,19].”

  1. Inclusion and exclusion criteria for selection of DPT need to be added.

We have included additional criteria for sample selection:

“The inclusion criteria were healthy children of Hispanic ethnicity without anomalies that might affect dental development. The exclusion criteria were panoramic radiographs of poor diagnostic quality that prevents scoring the tooth developmental stages.”

3.Comparrison of the findings with other ethnic groups and populations section need to be expanded as this will increase the validity of the study.

We have included additional studies to compare the outcomes of dental age estimation.

“The overall difference between the chronological and dental age (CA-DA) in the Hispanic children aged 7 to 14 years in the current study was -0.10 years for females (less than one week) and 0.03 years (less than one week) for males. This is similar to the result of a recent study on Hispanic children in Texas where a difference of 0.07 years for females and 0.03 years for males was reported on children aged 6 to 17 years [19]. This outcome could be also compared with other studies that utilized the same Dental Age Research London Information Group (DARLInG) methodology for the construction and validation of the reference dataset. For example, in the Kuwaiti population, it was 0.14 years for females, and -0.33 years for males [30], 0.03 years for females and 0.05 years for males in southern Chinese [10], and 0.03 years for females and 0.05 years for males Afro-Trinidadians [31]. It is to be noted that the age of subjects included in the above studies were between 2 and 25 years and may not allow direct comparison based on the age of children included in the current study.”

Reviewer 2 Report

Dear Authors, your manuscript can be considered as a nearly finished scientific paper contributing to forensic dentistry. I have just some minor recommendations and remarks highlighted in attached PDF.

This paper presents dental reference data set estimating the age of children between 8 to 12 years in California of self-reported Hispanic origin. Paper concludes that difference between the chronological and dental age estimated from the Hispanic reference dataset was found to be minimal within a range of few weeks.

I might have overlooked; however, it remains unclear to me your decision of addressing the age interval for reference dataset, from 7.00 to 13.99 years. Why did you apply limitation by the age of 14 years, and not covering all teenagers? This remains unclear/unexplained in the manuscript. Your consideration of expanding this study to include older children is reasonable. Assessing permanent tooth development for even younger Hispanic children within the reference data set would have allowed a more comprehensive representation of the variation in dental development.

Regarding cross-border migrations, an evaluation of differences in CA and DA estimations in children older than thirteen years of age would be very useful.

From my point of view the dental/chronological age evaluations will soon become a domain of AI algorithms, which will prevail in x-ray evaluations and will probably achieve higher accuracy than human observers. Many AI applications are already focused on this task, it might be worth mentioning this in the discussion. For example, Use of Advanced Artificial Intelligence in Forensic Medicine, Forensic Anthropology and Clinical Anatomy

In general, your manuscript requires a few minor syntax improvements, removal of Titles within the Abstract or expanding the keywords where words already used in the Title of the manuscript shall not be repeated.

Author Response

Reviewer 2

We thank the Reviewer for the comments. Please see below our response:

Dear Authors, your manuscript can be considered as a nearly finished scientific paper contributing to forensic dentistry. I have just some minor recommendations and remarks highlighted in attached PDF.

This paper presents dental reference data set estimating the age of children between 8 to 12 years in California of self-reported Hispanic origin. Paper concludes that difference between the chronological and dental age estimated from the Hispanic reference dataset was found to be minimal within a range of few weeks.

I might have overlooked; however, it remains unclear to me your decision of addressing the age interval for reference dataset, from 7.00 to 13.99 years. Why did you apply limitation by the age of 14 years, and not covering all teenagers? This remains unclear/unexplained in the manuscript. Your consideration of expanding this study to include older children is reasonable. Assessing permanent tooth development for even younger Hispanic children within the reference data set would have allowed a more comprehensive representation of the variation in dental development.

We understand this potential limitation. We have provided explanation and future recommendation:

“A limitation of this study is the age of children and adolescents included which is 7.00 to 13.99 years. This limits the application of age estimation within to this specific age range. Due to time constraint and lack of additional resources within the department, we could not include additional age ranges in the study. Future studies will include adolescents and young adults over 14 years which would make this dataset complete similar to other published studies [5,6,11,19].”

Regarding cross-border migrations, an evaluation of differences in CA and DA estimations in children older than thirteen years of age would be very useful.

Thank you. Kindly refer to our response on the above comment.

From my point of view the dental/chronological age evaluations will soon become a domain of AI algorithms, which will prevail in x-ray evaluations and will probably achieve higher accuracy than human observers. Many AI applications are already focused on this task, it might be worth mentioning this in the discussion. For example, Use of Advanced Artificial Intelligence in Forensic Medicine, Forensic Anthropology and Clinical Anatomy

Although this is an interesting suggestion, we feel that this is out of scope of the current study.

In general, your manuscript requires a few minor syntax improvements, removal of Titles within the Abstract or expanding the keywords where words already used in the Title of the manuscript shall not be repeated.

We have addressed minor syntax errors throughout the manuscript.

Reviewer 3 Report

In our current society, the issue of unaccompanied children seeking asylum is absolutely present and lack of personal documentation and falsification of age are serious problems and medico-legal issues.

From this perspective, the study is original and its contribution to forensic science and age estimation is undeniable.

Some minor revisions should be fixed by the authors:

-    The paragraph 3.1 Sample size should be reported under the second chapter Materials and methods.

-       English grammar language check should be required.

-    References can be further improved, since several papers have tried to investigate these issues in different population or using similar methodology (see Franceschetti et al., 2021 International Journal of Legal Medicine for example).

Overall, the potential of this technique is doubtless (the accessibility of dental panoramic radiograph and its non-invasiveness) and the authors have to expand this study to teenagers aged 14 – 18 years old and older in a further research.

Author Response

Reviewer 3

We thank the Reviewer for the comments. Please see below our response:

In our current society, the issue of unaccompanied children seeking asylum is absolutely present and lack of personal documentation and falsification of age are serious problems and medico-legal issues. 

From this perspective, the study is original and its contribution to forensic science and age estimation is undeniable.

Some minor revisions should be fixed by the authors: 

The paragraph 3.1 Sample size should be reported under the second chapter Materials and methods

We have deleted “3.1. Sample Size” in the Results and moved the contents to the Methods “2.1. Sample Population”.

English grammar language check should be required. 

We have checked English grammar language and made changes where appropriate.

References can be further improved, since several papers have tried to investigate these issues in different population or using similar methodology (see Franceschetti et al., 2021 International Journal of Legal Medicine for example). 

We have included additional studies and compared the outcomes. See “Discussion” section:

“The overall difference between the chronological and dental age (CA-DA) in the Hispanic children aged 7 to 14 years in the current study was -0.10 years for females and 0.03 years for males. This is similar to the result of a recent study on Hispanic children in Texas where a difference of 0.07 years for females and 0.03 years for males was reported on children aged 6 to 17 years [19]. This outcome could be also compared with other studies that utilized the same methodology for the construction and validation of the reference dataset. For example, in the Kuwaiti population, it was 0.14 years for females, and -0.33 years for males [11], 0.03 years for females and 0.05 years for males in southern Chinese [10], and 0.03 years for females and 0.05 years for males Afro-Trinidadians [6]. It is to be noted that the age of subjects included in the above studies were between 2 and 25 years and may not allow direct comparison based on the age of children included in the current study.”

Overall, the potential of this technique is doubtless (the accessibility of dental panoramic radiograph and its non-invasiveness) and the authors have to expand this study to teenagers aged 14 – 18 years old and older in a further research.

This is a good suggestion. We have provided explanation for the limited age range as a limitation and recommendations for future research in the “Discussion” section:

“A limitation of this study is the age of children and adolescents included which is 7.00 to 13.99 years. This limits the application of age estimation within to this specific age range. Due to time constraint and lack of additional resources within the department, we could not include additional age ranges in the study. Future studies will include adolescents and young adults over 14 years which would make this dataset complete similar to other published studies [5,6,11,19].”
